# Efficient Parallel Samplers for Recurrent-Depth Models and Their Connection to Diffusion Language Models

## Abstract

Language models with recurrent depth, also referred to as universal or looped when considering transformers, are defined by the capacity to increase their computation through the repetition of layers. Recent efforts in pretraining have demonstrated that these architectures can scale to modern language modeling tasks while exhibiting advantages in reasoning tasks. In this work, we examine the relationship between recurrent-depth models and diffusion language models. Building on their similarities, we develop a new diffusion forcing sampler for these models to accelerate generation. The sampler advances by decoding new tokens at every forward pass of the model, while the latent states of these tokens can be further refined in parallel through recurrence. Theoretically, generation with our sampler is strictly more expressive than the baseline autoregressive generation using the same time budget on modern hardware. Moreover, this sampler, based on principles from diffusion literature, can be directly applied to existing 3.5B recurrent-depth transformers without any tuning, leading to up to a **5**x speedup. Consequently, our findings not only provide an efficient mechanism for parallelizing the extra computation in recurrent-depth models at inference, but also suggest that such models can be naturally viewed as strong continuous, though causal, diffusion language models.

## 1 Introduction

Conventional large language models (LLMs) are constructed as fixed-depth neural networks with a predetermined number of layers (often merely a two-digit count), a property that not only allows these models to be trained efficiently, but in practice appears sufficient for many tasks (Radford et al., 2019). However, more challenging tasks in mathematics and programming often require conceptual leaps over multiple steps in a logical chain that are hard for these models to learn robustly. More formally, fixed-depth transformers fall within the complexity class $\mathsf{TC}^0$ (Merrill & Sabharwal, 2023). To resolve this, recent efforts have focused on training models to "verbalize" their internal reasoning into chains-of-thought composed of small sub-steps, each of which the model is capable of learning.

An alternative to fixed-depth are models with *recurrent depth* (Dehghani et al., 2019; Schwarzschild et al., 2021), which can repeat layers. Consequently, these models are also referred to as *looped* transformers (Giannou et al., 2023), or, as *universal* transformers, (Dehghani et al., 2019) when highlighting the motivation for these systems to represent universal Turing machines (Graves et al., 2014; Graves, 2017). Merrill & Sabharwal (2025) showcase that, in contrast to fixed-depth models, models with arbitrary recurrence are indeed capable of representing a larger complexity class.

However, generation with autoregressive recurrent-depth models is typically slow, given that every repetition of the model layers must be executed sequentially before the next token can be produced. In this work, we discuss how generation from recurrent-depth models can be efficiently parallelized by connecting this architecture to diffusion model architectures. Both architectures "recur" in a related sense, and even though both are trained with different objectives, we show that samplers adapted from diffusion literature, namely, *diffusion forcing* (Chen et al., 2024b), can be directly applied to parallelize the generation of already existing recurrent-depth models from Geiping et al. (2025).

We discuss how to adapt diffusion forcing sampling to recurrent-depth models, identifying the essential architectural components and strategies required to ensure both stability of the iterates and

**Figure 1:** Different generation schemes for autoregressive, recurrent-depth models. **Left:** Standard sequential generation, which proceeds one token and step of the recurrence at a time (time steps denoted by integers). **Right:** A diffusion forcing sampler used for the same model can parallelize generation "diagonally", by computing one step of the recurrence per token position, iteratively refining its estimate of the generated sequence.

bounded memory usage. As illustrated in Figure 1, rather than waiting for the recurrence at sequence position $n$ to fully converge before generating the next token, our sampler immediately produces token drafts from intermediate iterates. It then advances to position $n + 1$, where the subsequent forward pass simultaneously refines the drafts for steps $n$ and $n + 1$, while also decoding an initial draft for $n + 2$. In this way, the sampler achieves parallelism along the sequence dimension, akin to speculative decoding. Importantly, because the underlying model is trained as a causal language model, information still propagates strictly from left to right, and the output sequence is iteratively refined across recurrences. This parallelization is not guaranteed to generate the same solution, but we show in this work theoretically and practically that it generally leads to solutions with similar quality that are generated much quicker. While the approach does not reduce FLOPs, it effectively exploits modern GPU architectures by unlocking additional opportunities for parallelization. Overall, in this work, we

- Clarify the connection between recurrent-depth models and diffusion models via diffusion forcing and block or wave-based inference strategies for sequence-based diffusion models.
- Describe how to apply principles from diffusion forcing to efficiently parallelize the inference of models with recurrent depth.
- Verify that recurrent-depth models equipped with diffusion-forcing samplers achieve the strongest balance between practical efficiency and theoretical expressiveness in both prefilling and decoding.
- Show that diffusion forcing sampling outperforms even well-tuned speculative decoding baselines for the same model with speed gains that can be smoothly traded off against response quality.

## 2 RELATED WORK

We briefly introduce both recurrent models and diffusion models, focusing on language applications.

**Recurrent Models.** Models with recurrent computations have long been central to machine learning (Amari, 1972; Hopfield, 1982; Braitenberg, 1986; Gers & Schmidhuber, 2000; Sutskever et al., 2008), not only due to significant inspiration from recurrent firing patterns found in neuroscience (Hopfield, 1982; Lamme & Roelfsema, 2000; Douglas & Martin, 2004), and early successes in language modeling centered on recurrent neural networks (Mikolov et al., 2010; Sutskever et al., 2011). With the advent of transformer models, these architectures were considered less scalable, yet recurrence, now as *recurrence in depth*, was swiftly re-introduced as *universal transformers*, Dehghani et al. (2019), motivating that these models could be capable of modeling universal Turing machines (Graves et al., 2014). Other work showed that recurrent models were capable of learning algorithms (Schwarzschild et al., 2021; Bansal et al., 2022; Bear et al., 2024). That recurrence was capable of representing universal computation was explicitly constructed for transformer models in Giannou et al. (2023), and following work on *looped* transformers has shown that these models are capable learners (Giannou et al., 2023; Gatmiry et al., 2024; Yang et al., 2024; McLeish et al., 2024; Fan et al., 2025). These findings have led to a wave of work training larger, general-purpose recurrent-depth models of language (Tan et al., 2023; Abnar et al., 2023; Mathur et al., 2024; Csordás et al., 2024; Geiping et al., 2025), as well as work retro-fitting recurrence into trained models (Li et al., 2020; Bae et al., 2024; Hay & Wolf, 2023; Liu et al., 2024b). Several of these works also highlight the possibility of implementing *latent reasoning* via recurrence, that is to complement or replace verbalized chains-of-thought, with recurrence. Examples for this line of thinking are *Coconut* (Hao et al., 2024), as well as Liu et al. (2024a); Cheng & Durme (2024).

| t=3 | To determine the number I |
| t=4 | To determine how number of would |
| t=5 | To determine how many of eggs- |
| t=6 | To determine how many dozens dozens Claireto |
| t=7 | To determine how many dozens of of willons |
| t=8 | To determine how many dozens of eggs eggs be of |
| t=9 | To determine how many dozens of eggs Claire Claire Claire a |
| t=10 | To determine how many dozens of eggs Claire will will will day |
| t=11 | To determine how many dozens of eggs Claire will eat eat eat in |
| t=12 | To determine how many dozens of eggs Claire will eat in in in 4 |
| t=13 | To determine how many dozens of eggs Claire will eat in 4 4 4 weeks |

■ Frozen tokens, committed to KV cache   ■ Current Candidate Tokens   ■ Newly Sampled Token

**Figure 2:** An example of a text sequence being generated with the proposed diffusion forcing sampler from a depth-recurrent model. While the original recurrent-depth model requires 32 recurrence steps to produce a single token (the default for this model), the diffusion sampler has already produced and committed 8 new tokens (green). As described, the sampler advances by at least one token per step of the recurrence. Decoded candidate tokens are initial spell out incoherent text, but map into the right concepts, and quickly improve with more steps. Note that the "freeze" decision is dynamic, based on distance to the previous state in latent space (not pictured).

In this work, we propose a generic sampling algorithm for depth-recurrent models, which we test with the models developed in Geiping et al. (2025), which are trained for general language understanding and reasoning on 800B tokens, with 3.5B parameters, and openly accessible.

**Diffusion Language Models.** Diffusion models are general-purpose generative models, with early applications focusing on continuous domains, such as images (Song & Ermon, 2019; Rombach et al., 2022; Peebles & Xie, 2023), which lead to substantial interest in extending diffusion also to discrete domains, such as text (Austin et al., 2021; Hoogeboom et al., 2021). Approaches to language diffusion are split on whether to incorporate diffusion processes on a continuous variable (that is then projected into discrete space) (Chen et al., 2022; Dieleman et al., 2022; Han et al., 2023; Karimi Mahabadi et al., 2024; Jo & Hwang, 2025; Graves et al., 2025), or diffusion processes that directly act on discrete variables.(Lou et al., 2024; Richemond et al., 2023). The latter though, especially using *masking* as the discrete forward diffusion step, is currently the most scalable approach, employed in large-scale efforts to train language diffusion models, competitive with autoregressive models (Gong et al., 2025a;b; DeepMind, 2025; Nie et al., 2025; Wang et al., 2025b; Xie et al., 2025; Ye et al., 2025).

**Inference Strategies for Diffusion Language Models.** To make diffusion tractable for arbitrarily long sequences requires techniques such as block diffusion (Arriola et al., 2025), where chunks of text are being modified by the diffusion model, and then frozen and their KV entries cached, with the sampler moving to the next chunk. A more free-form approach to handle sequence-based diffusion is to use *diffusion forcing* (Chen et al., 2024b), a hybrid model, where noise is added to future tokens in a sequence relative to the position of the current token, allowing the sampler to move both on both the sequence dimension and the diffusion time dimension.

**Inference Acceleration for Fixed-Depth Transformers.** Inference in transformers, in particular in small-batch settings is memory-bound, meaning that the transfer of data (or, in the default case, model parameters) to and from the L1 cache of the accelerator, is the dominating cost during inference, allowing algorithms such as speculative decoding (Leviathan et al., 2023) and follow-ups (Cai et al., 2024; Miao et al., 2024; Chen et al., 2024c) to improve inference speed through speculative parallelization. Using smaller draft models, these algorithms draft text several tokens in the future, which can then be verified using the original model, as verification of the entire text sequence is compute-bound and hence, fast.

## 3 APPLYING DIFFUSION FORCING TO RECURRENT-DEPTH MODELS

In this section, we present our diffusion forcing sampler for recurrent-depth models, which accelerates text generation by advancing at least one token in each recurrence step, as illustrated in Figure 2.

### 3.1 UNDERSTANDING RECURRENT-DEPTH AS LATENT DIFFUSION

Before detailing the diffusion forcing sampler, we briefly describe the particular recurrent-depth architecture proposed by Geiping et al. (2025), emphasizing features of the model that are pertinent to the

sampler's functionality. We will use the checkpoint name *Huginn-0125* when referring to the trained model. The architecture of this model contains three main blocks, each composed of multiple transformer layers: (i) a prelude block $P$, projecting the embedded input tokens into a latent space; (ii) a recurrent block $R$, iterating $r$ times in this latent space by refining a state vector $\mathbf{s}$, and (iii) a coda block $C$ that processes the latent state and produces the model's probabilities for the next token, formally

$$\mathbf{e} = P(\mathbf{x})$$
$$\mathbf{s}_0 \sim \mathcal{N}(\mathbf{0}, \sigma^2 I)$$
$$\mathbf{s}_i = R(\mathbf{e}, \mathbf{s}_{i-1}) \qquad \text{for} \quad i \in \{1, \dots, r\}$$
$$\mathbf{p} = C(\mathbf{s}_r).$$

Notably, while this architecture is derived from looping the middle layers of fixed-depth transformer models (Skean et al., 2024; Sun et al., 2024; Kaplan et al., 2024), with features such as input injection and random state initialization from the literature of recurrent-depth models (Bansal et al., 2022; Anil et al., 2022), it can also be interpreted as a *latent-space diffusion model* following the formulation of Rombach et al. (2022): Starting from an initial random state $s_0$, the model iteratively refines this state conditioned on the embedded input sequence $e$, until we assume the state to be completely denoised at the end of the process, at which point it will be decoded into the next token using $C$.

In Geiping et al. (2025), this model is trained using randomized unrolling with truncated backpropagation, i.e. a random number of iterates $r$ is sampled (from a Poisson-lognormal distribution), and then the entire current batch of training sequences is iterated up to $r$, which is not directly related to diffusion language modeling, which most effectively trains by randomized masking and adaptation from autoregressive models (Nie et al., 2025; Xie et al., 2025; Ye et al., 2025; Gong et al., 2025a).

## 3.2 THE INGREDIENTS FOR DIFFUSION FORCING SAMPLING

While we will describe experiments using this particular recurrent-depth model, the sampler can be applied to all recurrent-depth models that fulfill the following requirements.

**Input Injection.** The first necessary component, aside from the recurrence over layers itself, is the input injection, i.e., the conditioning of the recurrence on $e$. This will allow the sampler to "course-correct" if conditioning changes without having to jettison a partially computed state $s$. The other component that may improve the connection to diffusion modeling is the initialization of random states, but while we speculate that this is beneficial, it is not architecturally necessary. As such, recurrent-depth models trained in Csordás et al. (2024); Schöne et al. (2025); Mohtashami et al. (2024) or Wang et al. (2025a) could also benefit from this sampler. However, looped architectures such as *Coconut* (Hao et al., 2024), which train to feed the outputs of a transformer back in as inputs, are not immediately supported and require retraining to incorporate input injection, separating their recurrent state from their input data.

**Robust Recurrence.** The second necessary property is that the intermediate state at every step of the recurrence must be decodable to approximately correct solutions. While this property is generally satisfied, it may fail in models trained exclusively with a fixed number of recurrences $r$, where decoding from earlier steps can yield nonsensical outputs rather than approximate versions of the intended result.

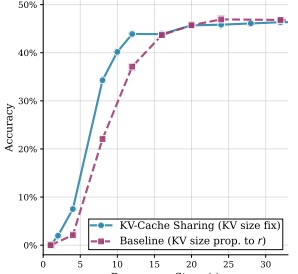

**Figure 3:** The *Huginn-0125* recurrent-depth model can match the baseline performance on the GSM8k dataset when enabling KV cache sharing (with a minimal cache size of 1), using $r$-times less memory for KV states.

**KV Cache Sharing.** The third property, while not strictly required but highly beneficial for diffusion forcing samplers, is the ability of different recurrent depths to share their KV cache across iterations during generation. Without fungible KV states, all KV states from previous recurrences and tokens must be retained in memory, causing the cache to grow with both sequence length and recurrence depth. As shown in Figure 3, the trained *Huginn-0125* model inherently supports KV cache sharing, allowing us to store only the KV state of the most recent recurrence for each token position[1].

---

[1]With this form of KV sharing, the cache requires no more memory than that of a parameter-matched fixed-depth transformer.

---

**Algorithm 1** Diffusion-forcing-style generation, simplified version (Full Version in Algorithm 2)

---

**Require:** current text context $\mathbf{x}$, max new tokens $N$, inner recurrence $r'$, total recurrences per token $r$, diffusion steps $T$, init scale $\alpha$

1:  $\mathbf{y}_{\text{frozen}} \leftarrow \mathbf{x}$
2:  $\mathbf{y}_{\text{current}} \leftarrow \mathbf{x}$
3:  $\mathbf{z} \leftarrow \text{InitState}(1, \alpha)$
4:  **for** step $t = 1, \ldots, T$ **do**
5:     $\mathbf{e} \leftarrow \mathcal{P}(\mathbf{y}_{\text{current}})$
6:     $\mathbf{z}_{\text{noise}} \leftarrow \text{InitState}(1, \alpha)$
7:     $\mathbf{z} \leftarrow (1 - \beta_t)\mathbf{z} + \beta_t\, \mathbf{z}_{\text{noise}}$
8:     **for** $j = 1, \ldots, r'$ **do**
9:        $\mathbf{z} \leftarrow \mathcal{R}(\mathbf{z}, \mathbf{e})$                           $\triangleright$ Inner recurrence
10:    **end for**
11:    $\mathbf{p} \leftarrow \mathcal{C}(\mathbf{z})$                                  $\triangleright$ project latent states to logits
12:    $\hat{\mathbf{y}} \leftarrow \text{Sample}(\mathbf{p})$
13:    $\mathbf{y}_{\text{current}} \leftarrow [\mathbf{y}_{\text{frozen}}, \hat{\mathbf{y}}]$
14:    $\mathbf{y}_{\text{frozen}} \leftarrow \text{Assign } \mathbf{y}_{\text{current}} \text{ up to the last } \lceil \frac{r}{r'} \rceil \text{ entries}$      $\triangleright$ Freeze completed tokens
15:    **if** $|\mathbf{y}_{\text{frozen}}| - |\mathbf{x}| \geq N$ **then break**
16:    **end if**
17:    $\mathbf{z} \leftarrow [\mathbf{z}, \text{InitState}(1, \alpha)]$         $\triangleright$ Append a new latent state for the next position
18: **end for**
19: **return** $\mathbf{y}_{\text{frozen}}$

---

### 3.3 A Simplified Version of the Sampling Algorithm

Next, we present the algorithm for our sampler. Given a prompt $x$, Algorithm 1 describes a simplified version that directly adapts diffusion forcing principles to parallelize generation across the sequence dimension. This approach yields improvements in tokens/second while maintaining equivalent total FLOP requirements. An example of the sampler's behavior is illustrated in Figure 2.

We emphasize several important aspects. First, the number of inner recurrences $r'$ may be chosen to exceed one. These additional iterations are relatively inexpensive, since the broader logic of the sampler is not yet invoked. More importantly, they serve to stabilize the recurrence. Because the conditioning on the input embedding $\mathbf{e}$ may vary across successive steps of the sampler, the model risks becoming trapped in oscillatory behavior unless sufficient steps are allowed to adapt the current state to the evolving conditioning. This mechanism closely parallels practices in the diffusion literature, such as the use of supplementary diffusion steps in Bansal et al. (2023) to incorporate complex guidance signals into image diffusion models.

Second, we naturally employ this sampler only during the generation phase, as the prefill phase is already parallelizable in the sequence dimension, as the recurrence can be computed on all token positions of the prompt simultaneously.

Further, in terms of efficiency, we note that we do not actually want to keep the state for all tokens changing indefinitely, as doing so would slow down generation again, as well as increase memory usage dramatically. As such, similar to block-diffusion samplers (Arriola et al., 2025), we look for rules that decide when each position is "finished". In the simplified version of the sampler, we freeze the last token once we reach a predetermined number of recurrence steps at this position – which naturally happens $r$ positions behind the current maximal extent of the sequence. Frozen tokens are removed from the state vector and their KV states are added to the cache, so that, as in block diffusion models (Arriola et al., 2025), at each point in time, only a small subset of tokens in being modified and the full generation runs like a wave over the generating sequence. Finally, note that with this simplified exit rule, $r' = r$ exactly recovers the original autoregressive sampler.

### 3.4 Stabilizing components based on Diffusion Principles

Further, we also experiment with adding momentum to the input conditioning $\mathbf{e}$, setting

$$\mathbf{e} = \eta\, \mathbf{e}_{\text{prev}} + (1 - \eta)\mathcal{P}(y_{\text{current}}), \tag{1}$$

which we find can stabilize the recurrence in challenging sequences, providing a small, but robust gain on average.

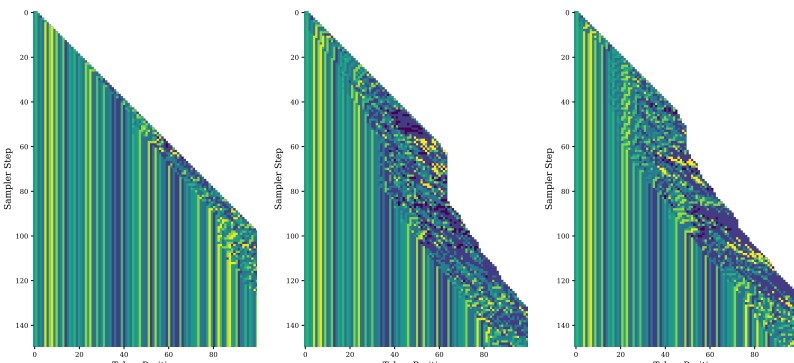

**Figure 4: Examples of adaptive sampler behavior**. Each color represents a token id in the vocabulary of the model, showing the development of the generated sequence (running left to right) as a function of sampler steps (running top to bottom) for *different hyperparameter choices*. The leftmost example is $r' = 4$, and tokens are frozen quickly, whereas middle and right show sequences with $r < 4$ require more adaptive computation, and in both cases the sampler stalls after hitting the maximal length of the wavefront (here 32 to visualize), before resolving the sequence and advancing again.

Secondly, surprisingly, we find that even though these models are never trained with noise injected into intermediate states, that artificially adding noise to the state in each step of the sampler, in analogy to sampling from continuous diffusion models, i.e.

$$\mathbf{z}' = (1 - \beta_t)\mathbf{z} + \beta_t \mathbf{z}_{\text{noise}} \qquad \text{where} \quad \mathbf{z}_{\text{noise}} = \text{InitState}(1, \alpha), \tag{2}$$

can stabilize the iterative process, leading to gains in both accuracy and throughput if $r'$ is small. In practice, we schedule $\beta_t$ linearly as a function of steps $t$ at each position, so that the latter steps are naturally less noisy (Chen et al., 2024a), which we find to outperform either scheduling $\beta_t$ scaled by the square root of the number of recurrences at each position or keeping it constant. However, the optimal value of $\beta_t$ depends on $r'$.

### 3.5 ADAPTIVE EXITS

However, the fixed exit scheme of the simplified sampler can run into issues. The recurrent-depth model is causal and how quickly states converge depends on the complexity of the query. This can lead to situations where either, compute is wasted because the states at certain positions have already converged quicker than $r$, or, more problematically, states where, due to a late change in the conditioning of prior tokens, the states have not converged in time. Freezing these unfinished states would worsen generation, in the worst case leading to a spiral where each token that is frozen incorrectly slows down convergence further, leading to a response that becomes more incorrect with each token.

However, we can remedy both cases through adaptive compute. We pick the simplest adaptive exit criterion, the normalized distance in latent space, and compute this quantity for each position and freeze up to all positions where this distance $\delta_i$ is smaller than a threshold $\varepsilon$.

$$\delta_i = \frac{\|\mathbf{z}_i - \mathbf{z}_{\text{prev},i}\|_2}{\|\mathbf{z}_i\|_2}, \qquad k^* = \max\{k : \delta_j < \varepsilon \text{ for all } j \leq k\} \tag{3}$$

We combine this with a limiter on the maximum length of the wavefront of the algorithm to guarantee that both 1) the number of states currently being modified, so the maximum memory footprint, is bounded and 2) only positions with converged states are frozen. The full algorithm is described in Appendix Algorithm 2. With these rules in place, we note that setting the wavefront to 1 token, we exactly recover the token-per-token adaptive compute sampler from (Geiping et al., 2025).

We show the practical outcome of this sampler for a challenging input sequence from GSM8k in a series of heatmaps in the appendix, see Figure 13. The heatmap shows the development of the sequence as a function of generation steps and tokens. We see that the wave first advances quickly, but then halts for a short amount of steps, before resuming the advance.

**Remark 3.1** (Convergence of the Adaptive Diffusion Sampler). *With this algorithm, we can, in principle guarantee convergence to the same solution as when sampling autoregressively, if we assume that the recurrent block $R$ was a contraction. Then, convergence of iterates, i.e. Equation (3), would imply convergence to the fixed point of the operator. Second, because the model is causal,*

*convergence of the first token position does not depend others and will converge at some step $t$. At this step, the conditioning of the subsequent token is frozen, so it will also converge, proving convergence of the full sequence to the autoregressive solution by induction. However, in practice, large-scale recurrent-depth models are not easily proven to be contractive, even if models are approximately path-independent (Anil et al., 2022), so we provide this only as a conceptual remark.*

## 4 THEORETICAL ANALYSIS

This section develops a theoretical framework to justify the optimality of our design in balancing efficiency and expressiveness with two research questions (RQs): **(i)** Why should models prioritize recurrence, i.e. *depth scaling*, during prefilling? and **(ii)** Why should models prioritize parallelizing decoding from a larger wavefront of tokens using the sampler described in the previous section, i.e. *width scaling* during decoding?

### 4.1 PROBLEM FORMULATION

Before answering these RQs, we formalize the notions of depth and width within our framework, which limits our analysis to Transformer-based autoregressive LLMs. In particular, we focus exclusively on the comparison between depth and width, without considering length (i.e., CoT) scaling.

**Definition 4.1** (Depth and Width in Recurrent-Depth Models, informal)**.** For recurrent-depth models, we define *depth* $d_t$ and *width* $w_t$ at each time step $t \in \mathbb{N}$, with initial conditions $d_0 = 0$ and $w_0 = L_0$ (where $L_0$ denotes the input sequence length). The corresponding update rules are given as follows:

1. **Depth Update:** At each step $t$, $d_{t+1} = d_t + 1$ with $d_0 = 0$, therefore $d_t = t$ for all $t \in \mathbb{N}$.

2. **Width Update:** At each step $t$, width changes only through token exits and token entries:

$$\delta^{(t)} = \begin{cases} -1, & \text{if a hidden state decodes from the model (exit event),} \\ +1, & \text{if a latest token encodes into the model (entry event).} \end{cases}$$

### 4.2 LLMS SHOULD PRIORITIZE DEPTH SCALING DURING PREFILLING.

To establish this, we first define a width scaling architecture without increasing model parameters following Wu et al. (2025). Concretely, we repeat each token along the sequence dimension. Note that during prefilling, increasing the number of such repeated tokens is equivalent to width scaling under our definition, since this expands the input sequence length. Here, we introduce two variants:

- Width Scaling without KV Sharing (Width-NoShare): For the $j$-th copy of token $i$, attention is allowed to all copies of tokens $0, \ldots, i-1$, as well as the first $j-1$ copies of token $i$.
- Width Scaling with KV Sharing (Width-KVShare): For the $j$-th copy of token $i$, attention is limited to (i) the last copy of tokens $0, \ldots, i-1$, and (ii) the first $j-1$ copies of token $i$.

Based on the above definition, we state the importance of depth scaling during prefilling stage.

**Theorem 4.2** (Depth vs. Width Scaling in Prefilling, informal)**.** *Given the width-scaling architecture above and our recurrent-depth model with the same scaling factor $s$. Then the following hold:*

*1. **Expressiveness.** Under equal scaling factors, depth scaling is more expressive than width scaling.*

*2. **Complexity.** For asymptotic prefill cost (including both attention and linear layers), we have*

$$E_{\text{Depth}} \leq E_{\text{Width-KVShare}} < E_{\text{Width-NoShare}}.$$

*3. **Parallelism.** There exists a threshold $L_\star$ such that for $L < L_\star$, width scaling provides $s^2$ times the parallelism of depth scaling, while for $L \geq L_\star$ both saturate with similar parallelism.*

**Remark 4.3.** *Let $L$ be a random variable for prompt length with distribution $\mathcal{D}$. Then the probability that depth scaling is more efficient than width scaling equals $\Pr_{L \sim \mathcal{D}}[L \geq L_\star]$. Since $L_\star$ on modern GPUs typically lies between a few hundred and a few thousand tokens while empirical input length distributions place substantial mass above this range, the probability is indeed close to $1$ in practice.*

| Sampler | GSM8K | | MATH500 | | HumanEval | | MBPP | |
|---|---|---|---|---|---|---|---|---|
| | Acc | t/s | Acc | t/s | Acc | t/s | Acc | t/s |
| Static AR ($r = 32$) | 41.77% | 36.1 | 17.60% | 6.4 | 22.56% | 13.5 | 31.60% | 15.3 |
| Static AR ($r = 4$) | 1.59% | 312.9 | 3.20% | 18.6 | 0.61% | 244.1 | 1.40% | 49.6 |
| Static AR ($r = 8$) | 31.61% | 137.5 | 14.80% | 23.1 | 21.34% | 61.7 | 27.40% | 57.2 |
| Static AR ($r = 64$) | 42.15% | 18.2 | 18.60% | 3.4 | 22.56% | 7.3 | 30.20% | 7.6 |
| Adaptive Compute AR | 42.23% | 66.9 | 18.20% | 12.2 | 21.95% | 26.1 | 30.20% | 29.5 |
| Speculative Decoding AR | 42.76% | 69.5 | 17.80% | 13.4 | 20.12% | 27.5 | 30.60% | 31.6 |
| Diff. Sampler ($r' = 2, \beta_t = 0.5$) | 40.71% | 182.2 | 17.60% | 35.9 | 20.12% | 67.4 | 27.80% | 92.3 |
| Diff. Sampler ($r' = 4, \beta_t = 0$) | 42.08% | 157.3 | 18.00% | 30.3 | 20.12% | 64.9 | 31.00% | 70.2 |
| Relative Diff to AR ($r = 32$) | +0.31 | **4.36x** | +0.40 | **4.73x** | -2.44 | **4.81x** | -0.60 | **4.59x** |

**Table 1:** Performance comparison of autoregressive (AR) and diffusion samplers for the *Huginn-0125* model using a comparable backend (batch size 1, `transformers` with dynamic KV caching, no further inference optimizations). For both samplers, we record the total evaluation time divided by the number of samples. "Acc" denotes task accuracy, and "t/s" denotes the median of tokens/second measurements for all samples in the task.

### 4.3 LLMs SHOULD PRIORITIZE WIDTH SCALING DURING DECODING.

Using this framework, we can compute when recurrent-depth models should use diffusion forcing samplers during decoding.

**Theorem 4.4** (Depth vs. Width Scaling in Decoding, informal). *For recurrent-depth models with $r > 1$ inner recurrences, if diffusion forcing sampling and KV-cache sharing are employed with wavefront size $W \leq L_\star$, then diffusion forcing decoding achieves equal depth and strictly greater width compared to standard autoregressive decoding under the same runtime constraints. Mathematically, this relationship can be expressed as:*

$$d_{DF}(T) = d_{AR}(T) \quad and \quad w_{DF}(T) > w_{AR}(T),$$

*where $T$ is the runtime budget, and DF and AR denote diffusion forcing and autoregressive decoding.*

**Remark 4.5.** *Since model parameters and KV states are shared, the I/O cost of processing multiple tokens is asymptotically equivalent to processing a single token, enabling increased token generation within identical runtime constraints. At each decoding step, an expanded wavefront enables greater width scaling, providing superior expressiveness compared to autoregressive decoding. Empirically, since maximum recurrence depth rarely exceeds $r \approx 100$, the condition $W \leq L_\star$ typically holds.*

## 5 EXPERIMENTAL EVALUATION

To assess whether our method really accelerates generation, we compare our sampler against an equally optimized implementation of standard autoregressive sampling, both evaluated with a batch size of 1. Extensions to larger batch sizes are conceivable but fall outside the scope of this study, see additional discussion in Section A.2

We evaluate the 4 generative benchmarks (GSM8K, MATH500, HumanEval and MBPP) also evaluated in (Geiping et al., 2025), which we rerun using our sampler and compare against a number of baselines. Aside from the **static, autoregressive baseline** (static AR), at different recurrence steps, we also compare against the **adaptive compute** sampler of the original work, which still samples token-by-token, but exits the recurrence at every token, once the difference in latent space is small enough. We tune this sampler, finding that its hyperparameter, the threshold $\varepsilon$ is similar to the diffusion sampler. Finally, we also compare against a heavily tuned **self-speculative decoding baseline**. It was observed in Geiping et al. (2025) that recurrent-depth models can be natively used as their own draft models, using fewer steps to draft. We find that drafting 4 tokens into the future, each with 4 draft steps is optimal for the *Huginn-0125* checkpoint on GSM8k.

We implement all samplers in comparable Hugging Face `transformers` implementations with dynamic KV caching and we measure mean accuracy and median tokens per second, computed

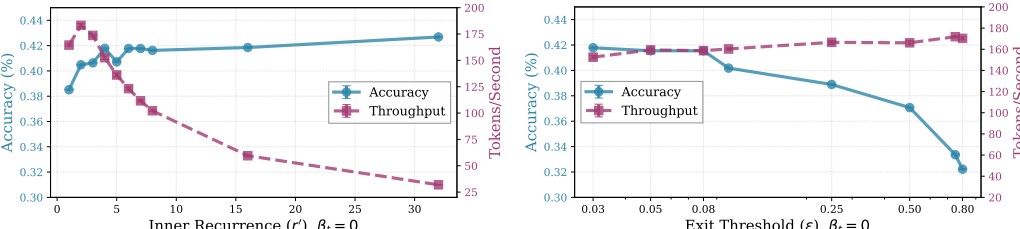

**Figure 5:** Trade-off between accuracy and speed on GSM8k under different hyperparameter choices. **Left:** Effect of increasing inner recurrence $r'$. Inner recurrence stabilizes the sampling, increasing accuracy at the cost of throughput. **Right:** Effect of varying the exit threshold $\varepsilon$. Modulating the exit threshold most directly trades off throughput and accuracy.

| Sampler | GSM8K | | Minerva Math | | HumanEval | | MBPP | |
|---|---|---|---|---|---|---|---|---|
| | Acc | Time | Acc | t/s | Acc | t/s | Acc | t/s |
| Huginn-0125 | | | | | | | | |
| Static AR ($r = 32$) | 41.77% | 36.1 | 12.98% | 21.0 | 22.56% | 13.5 | 31.60% | 15.3 |
| Diff. Sampler ($r' = 4, \beta_t = 0$) | 42.08% | 157.3 | 13.06% | 96.0 | 20.12% | 64.9 | 31.00% | 70.2 |
| SWA Model Variant | | | | | | | | |
| Static AR ($r = 32$) | 47.99% | 36.2 | 14.86% | 22.1 | 23.78% | 14.9 | 31.20% | 11.8 |
| Diff. Sampler ($r' = 4, \beta_t = 0$) | 47.08% | 143.1 | 14.52% | 101.4 | 23.78% | 71.2 | 29.20% | 59.7 |
| Math-Finetuned Model | | | | | | | | |
| Static AR ($r = 32$) | 58.91% | 29.8 | 22.20% | 7.9 | 17.07% | 11.5 | 28.80% | 11.2 |
| Diff. Sampler ($r' = 4, \beta_t = 0$) | 58.45% | 144.1 | 21.40% | 39.8 | 15.24% | 47.9 | 27.60% | 57.1 |

**Table 2:** Hyperparameters remain stable across different model variants. For example, both the weight-averaged checkpoint from the original work and the model finetuned on MetaMath for this study exhibit consistent speed gains in the range of 4–5×, and accuracy deviations within 0.5–1%, even when baseline values change.

over queries from each benchmark. All timings are obtained from CUDA event measurements on sandboxed A100-40GB GPUs. If not otherwise mentioned, we default to conservative settings for the sampler, always setting an exit threshold of $\varepsilon = 0.03$, $\beta_t = 0$, $\eta = 0.1$ and $r' = 4$, for a maximum wavefront size of $128$, if not otherwise mentioned.

**Benchmark Results.** We summarize our findings in Table 1. We find that on all benchmarks, executing the parallelized sampler leads to significant speedups of around 5x, with only minor trade-offs in generation quality of around 1%, depending on the task, owing to the trade-off set by our default hyperparameters. In Table 2 we repeat all benchmarks for two additional model checkpoints, the SWA model also released in Geiping et al. (2025), and a math variant, that we finetuned on the MetaMath dataset (Yu et al., 2023). Even though these model variants differ noticeably in their benchmark scores, they show similar gains and trade-offfs when using the diffusion sampler.

**Hyperparameter Choices.** We show the trade-off curves arising when varying the inner recurrence $r'$ and the exit threshold $\varepsilon$ in Figure 5 for two settings of noise $\beta_t$, finding that we can effectively trade-off additional generation speed against minor losses in accuracy. We further vary the embedding

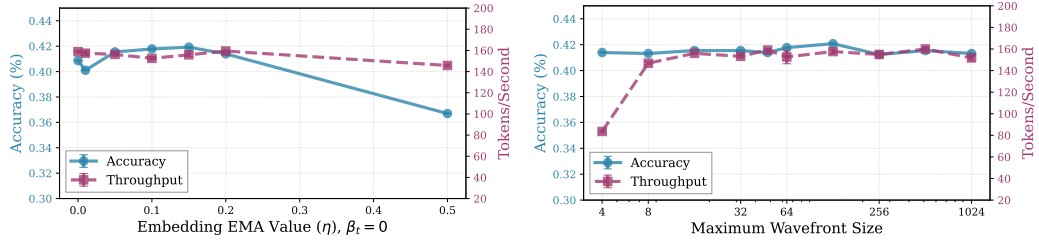

**Figure 6: Left:** Scaling the amount of momentum $\eta$ in the conditioning., showing that small, but non-zero $\eta$ values are optimal. **Right:** Size of the wavefront. Increasing wavefront size up to a value around 64-128 appears optimal. We note that the optimal wavefront size is also likely to be accelerator-specific.

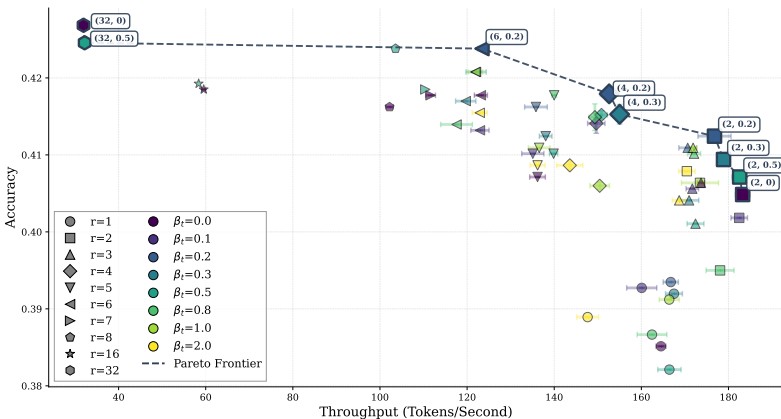

**Figure 7:** The Pareto Curve of Accuracy and Throughput on GSM8k spanned by varying inner recurrence and noise hyperparameter pairs $(r', \beta_t)$. Adding moderate amounts of noise, e.g. $\beta_t = 0.2$ is dominating runs with no noise added. Note also the scale of y-axis, as we are observing accuracy losses of only $2\%$ on the frontier.

| Dataset | Text Similarity to AR | | | LLM-as-Judge Evaluation | | | | | Log-Perplexity | | |
| --- | --- | --- | --- | --- | --- | --- | --- | --- | --- | --- | --- |
| | ROUGE | | Ident. Resp. | Prefer. (Win or Tie) | Correct | Helpful | Coherent | Complete | Self | Ext. (Qwen) | Tok/ Sec |
| | 1 | L | | | | | | | | | |
| **MTBench** | | | | | | | | | | | |
| AR Baseline | | | | | 4.31 | 4.75 | 5.83 | 5.34 | 0.82 | 1.42 | 8.04 |
| Diff. Sampler | 0.68 | 0.55 | 9% | 58% | 4.26 | 4.67 | 5.69 | 5.15 | 0.90 | 1.33 | 24.53 |
| **AlpacaEval** | | | | | | | | | | | |
| AR Baseline | | | | | 5.00 | 5.32 | 6.79 | 5.76 | 0.92 | 1.52 | 8.10 |
| Diff. Sampler | 0.68 | 0.55 | 11% | 57% | 4.88 | 5.15 | 6.43 | 5.42 | 0.95 | 1.56 | 26.48 |
| **LitBench** | | | | | | | | | | | |
| AR Baseline | | | | | 4.23 | 4.49 | 5.38 | 4.96 | 1.12 | 1.66 | 8.10 |
| Diff. Sampler | 0.50 | 0.24 | 0.6% | 44% | 3.91 | 4.17 | 4.82 | 4.08 | 1.17 | 1.72 | 26.48 |
| **Hermes-3** | | | | | | | | | | | |
| AR Baseline | | | | | 4.54 | 4.84 | 6.20 | 5.70 | 0.72 | 1.63 | 9.61 |
| Diff. Sampler | 0.75 | 0.67 | 31% | 71% | 4.67 | 4.82 | 6.22 | 5.50 | 0.69 | 1.63 | 34.37 |

**Table 3:** Comparison of free-form generation quality of autoregressive (AR) vs. Diffusion Sampling (Diff), covering textual overlap between both sampling strategies (in ROUGE F1 and percentage of identical answers), judge preference (measuring percentage of answers that tie or exceed the AR baseline) and rating (from 1-10), both via Claude Sonnet-4.5 and perplexity evaluated on the original model, and externally using Qwen-3-8b (base).

EMA $\eta$ and the noise schedule in Figure 6, showing that the sampler is robust to a broad range of settings for both options, although upsides are also limited. In Figure 7, we sweep a range of values for $r'$ and $\beta_t$, showing that, on average, more noise is helpful if the model takes fewer inner recurrence steps. In Figure 6 (right), we confirm that larger maximum wavefront sizes (i.e. the number of tokens that is modified at once in the adaptive sampler) allow for better parallelization. For the tested A100 GPU, the optimal maximal wavefront size is between 64 and 128, although this is likely accelerator-specific. In Table 3 we compare free-form generation quality metrics for a range of prompt datasets, measuring text overlap and LLM-as-a-judge ratings. We find that the speed trade-off is similar for free-form as for the reasoning benchmarks evaluated earlier.

## 6 CONCLUSIONS: ARE RECURRENT-DEPTH TRANSFORMERS SECRETLY CONTINUOUS LANGUAGE DIFFUSION MODELS?

We have shown that, surprisingly, diffusion forcing samplers can be directly applied to parallelize the inference of existing recurrent-depth language models, which we justify theoretically, and implement in practice, leading to five times faster single-sequence inference, even on reasoning and coding benchmark questions. Interestingly, we could also interpret this relationship in the opposite direction, namely that some recurrent-depth models *could* be thought of effectively as continuous latent language diffusion models, just trained with an unusual objective, namely truncated unrolling. This would imply that unrolling objectives could be competitive objectives for future language diffusion models.

## REPRODUCIBILITY STATEMENT

We provide a code submission with this submission that contains the complete sampler we describe, including all options that can be directly slotted into pre-existing recurrent-depth modeling code. We provide experimental details in Section 5 and provide further ablations and variants in the appendix. If not otherwise mentioned, all measured values are based on at least 5 repeated experiments. All timing are measured using CUDA events on GPUs of equal power, and are comparable to timings in the same table or figure.

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

# A APPENDIX

## A.1 ADDITIONAL ALGORITHM DETAILS

We provide the full algorithm, including adaptive exiting in Algorithm 2.

## A.2 ADDITIONAL VARIANTS AND CONSIDERATIONS

**What are the memory costs of this parallelization strategy?** The maximal memory allocated during generation is dynamic, given that token positions exit the current stack of states as their latent states converge and headway-many states are added per step of the sampler. However, the maximum wavefront size parameter (see also Figure 6 controls the maximum extent of the wavefront, thereby guaranteeing that the maximum memory allocation is bounded. In Figure 8 we provide additional details regarding the precise peak memory during generation as a function of maximal wavefront size. Plotted here is throughput as before on the y-axis, and, on the x-axis, we show the peak of additional memory allocated during the generation phase for a given sample. On the left side, the median additional memory peak is plotted, whereas on the right side, the worst-case event is shown. While for most samples, increasing the wavefront size does not necessarily increase memory usage after a certain point (left), as states usually exit quickly enough, with a wavefront size that is too large, there are samples (right) that would substantially increase memory requirements, which would be a concern if serving at scale. We recommend picking the smallest maximum wavefront size that saturates throughput on a per-accelerator basis, to prevent worst-case memory allocation.

---

**Algorithm 2** Diffusion-style generation with latent-diference-based freezing

---

**Require:** prompt $\mathbf{x}$, max new tokens $N$, inner recurrence $r$, diffusion steps $T$, init scale $\alpha$, exit threshold $\varepsilon$

1: $\mathbf{y}_{\text{frozen}} \leftarrow \mathbf{x}, \mathbf{y}_{\text{current}} \leftarrow \mathbf{x}$
2: $\mathbf{z} \leftarrow \text{InitState}(|\mathbf{x}|, \alpha)$
3: $\mathbf{z}_{\text{prev}} \leftarrow \mathbf{z}$
4: **for** step $t = 1, \ldots, T$ **do**
5:      $\mathbf{e} \leftarrow \mathcal{P}(\mathbf{y}_{\text{current}})$
6:      $\mathbf{z}_{\text{noise}} \sim \mathcal{N}(0, \sigma^2 I)$
7:      $\mathbf{z} \leftarrow (1 - \beta_r)\mathbf{z} + \beta_r \mathbf{z}_{\text{noise}}$
8:      **for** $j = 1, \ldots, r$ **do**
9:          $\mathbf{z} \leftarrow \mathcal{R}(\mathbf{z}, \mathbf{e})$
10:      **end for**
11:      $\mathbf{p} \leftarrow \mathcal{C}(\mathbf{z})$
12:      $\hat{\mathbf{y}} \leftarrow \text{Sample}(\mathbf{p})$
13:      $\mathbf{y}_{\text{current}} \leftarrow [\mathbf{y}_{\text{frozen}}, \hat{\mathbf{y}}]$
14:      $\delta_i \leftarrow ||\mathbf{z}_i - \mathbf{z}_{\text{prev},i}||_2 / ||\mathbf{z}_i||_2$          $\triangleright$ Compute relative changes in latents at each position.
15:      **if** exists position $i$ with $\delta_i < \varepsilon$ **then**
16:          let $k^* \leftarrow$ index of the last such freezable position where $\delta_i < \varepsilon$      $\triangleright$ freeze up to $k^*$
17:          $\mathbf{y}_{\text{frozen}} \leftarrow \mathbf{y}_{\text{current}}[1{:}k^*]$
18:          keep only unfrozen tail of latents: $\mathbf{z} \leftarrow \mathbf{z}[k^* - \ell{:}]$
19:      **else**
20:          no tokens frozen this step
21:      **end if**
22:      **if** $|\mathbf{y}_{\text{frozen}}| - |\mathbf{x}| \geq N$ **then break**
23:      **end if**
24:      $\mathbf{z} \leftarrow [\mathbf{z}, \text{InitState}(1, \alpha)]$          $\triangleright$ Append a new latent state for the next position
25:      $\mathbf{z}_{\text{prev}} \leftarrow \mathbf{z}$
26: **end for**
27: **return** $\mathbf{y}_{\text{frozen}}$

---

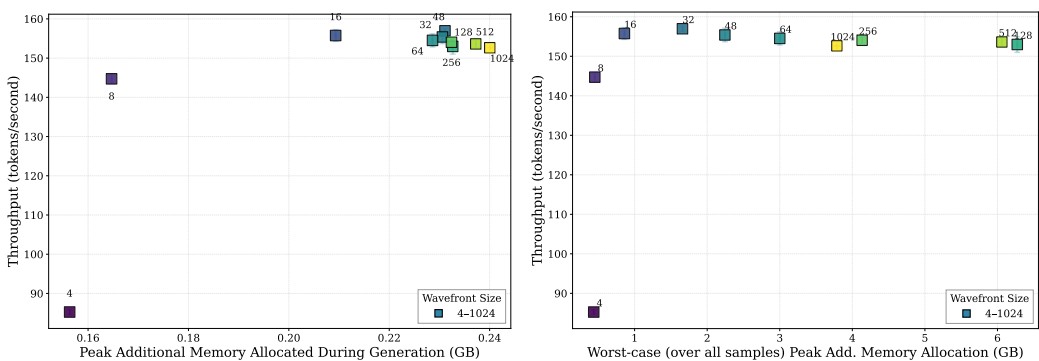

**Figure 8:** The median throughput in tokens/second and the median peak additional memory allocation during generation plotted as a function of maximum wavefront size, again for the *Huginn-0125* model on GSM8k. We can see that increasing the maximum wavefront size increases both throughput and memory, although after a certain point, the median memory increases only slightly (left). However, the max wavefront size does control the memory allocated to the worst-case sample (right), and we can see that the maximal memory allocated during the generation of that worst-case sample is indeed proportional to wavefront size (right).

**Moving Forward Multiple Steps.** In principle, there is no limitation of only advancing one token at a time, and so we can consider *headways* greater than 1, however, for these, we have no prior position to decode from, so we can only fill these positions with random tokens, or a particular padding token. And, given that the model is still causal, it will take several steps for sequential dependencies to be resolved, even if we sample a large headway in every step. We experiment with

headways greater than one, but while interestingly stable, this accelerates the speed of the sampler only marginally at a cost to accuracy, see Figure 9, right.

**Larger Batch Sizes.** The sampler discussed in this work could, in principle, also be deployed in batched or continuously-batched inference settings. In that scenario, similar to a paged KV cache, the sampler would reserve a number of slots for hidden states up to an occupancy multiplier of the maximum wavefront size, and would be capable of scheduling recurrent updates in tandem with sequence updates. For larger models, this would, if implemented efficiently, actually simplify deployment, as recurrent states are fungible, and e.g. states could be evicted from one device, and then bundled into the next forward call of the model on a different device, as the slots of the model's hidden states do not have to correspond to contiguous sequences in either the sequence or the recurrence dimension. However, due to to the imminent complexity of such an inference engine, we refrained from engaging with this direction in this work, and focus only on properly bringing the general idea of diffusion sampling to recurrent-depth models, and leave a batched inference engine as a limitation, potentially motivating future work.

Also, we remark on practical back-of-the-envelope estimates of runtime cost:

**Remark A.1** (Computational Cost)**.** *In comparison to the baseline autoregressive sampling algorithm where the recurrence is computed one token at a time, there are two additional sources of computational cost, the cost to encode and decode latent states using $\mathcal{P}$ and $\mathcal{C}$, and the potential cost incurred if convergence is slower than in baseline due to cascading effects of tokens changing late, as seen in Figure 4 if the adaptive version is used. The first cost depends on the size of the recurrent block $\mathcal{R}$, relative to prelude and coda. For the model we study in this work this is disadvantageous as the FLOP costs for prelude and coda equal one pass through the recurrent block. We define the FLOP costs of one pass through $\mathcal{R}$ as $f$, ignoring attention, so that the FLOP costs of one iteration of the sampler is roughly $(r' + 1)f$. Then, the total FLOP costs of running the baseline algorithm for $w$ tokens are $(r + 1)fw$, compared to $(r + \frac{r}{r'})fw$ for the non-adaptive diffusion sampler. However, as we will see, this FLOP inefficiency is counteracted in practice by the parallelization gains obtained from the sampler.*

### A.3 ADDITIONAL EXPERIMENTAL DETAILS.

**Finetuned Math Model:** To verify that our findings are not limited to the particular model checkpoint we evaluate, and its capabilities, we finetune the original checkpoint for one epoch with a trapezoidal learning rate schedule with a peak learning rate of $5 \times 10^{-7}$ using the MetaMath dataset (Yu et al., 2023). As suggested in the original work, we train the model with randomized unrolling, we set a mean of $r = 32$ and sample $r$ from an Exponential distribution. As a sidenote, we remark that while we do train the full model, most of the gains can also be achieved by just finetuning the adapter component of the model that maps inputs and states into the recurrent block.

**Dataset Details.** When evaluating GSM8k, we always refer to the CoT version of the dataset, which we provide to the model with the 8 few-shot examples associated with this variant as in Touvron et al. (2023). We always score GSM8k using the flexible-extract metric, i.e. by matching the last number in the model response against the reference answer. For MATH500, we follow the format of DeepSeek-AI et al. (2025), while for Minerva Math, we follow the updated format established in the lm-eval harness. For both, we grade answers using *math-verify*. For MBPP and HumanEval, we grade these benchmarks as normal. During inference we sample with a temperature of 0.2 and top-p factor of 0.95 as in Geiping et al. (2025).

**Details on Free-Form Generation Results.** All results for the four free-form prompt datasets in Table 3 are using the "default" diffusion sampler settings of $r' = 4$ and $\beta_s = 0.2$, which we compare to the standard autoregressive sampling approach with $r = 32$. We take all (804) prompts from the AlpacaEval benchmark dataset (Li et al., 2023), all (80) prompts from the MTBench dataset (Zheng et al., 2023), and the first 500 LitBench training samples (Fein et al., 2025) and the first 100 Hermes-3 training samples (Teknium et al., 2025), to cover a wide range of generic instruction data, creative writing and technical questions. For each dataset and each prompt, we compare both sampling strategies head-on. We first measure ROUGE-1 and ROUGE-L overlap between the samples, and we record the fraction of prompts where response strings where entirely equal (which they are often for either short, or very clear answers, whereas e.g. for creative writing, the chance of generating the same story is very small). We then use Sonnet-4.5, in particular,

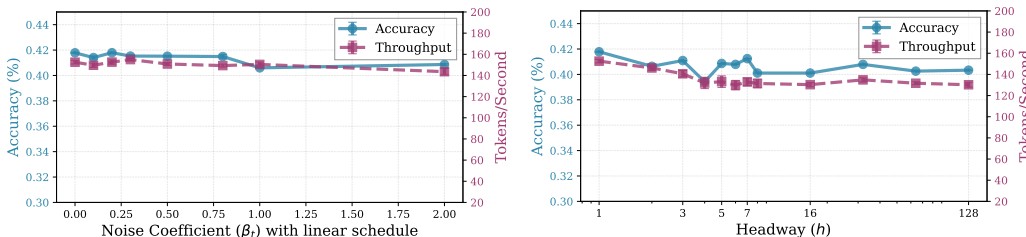

**Figure 9:** Impact of Additional Hyperparameter Choices on GSM8k. **Left:** Scaling the amount of noise added during inference for $r' = 4$, scheduled linearly in the number of recurrence steps, also measured on GSM8k. At $r' = 4$, adding noise is as impactful. We plot the full spectrum of $r'$ to $\beta_t$ in Figure 7. **Right:** Amount of headway. Larger amounts of headway than 1, i.e. advancing the sampler more than 1 token per step, do not seem to materialize practical speedups for the studied model.

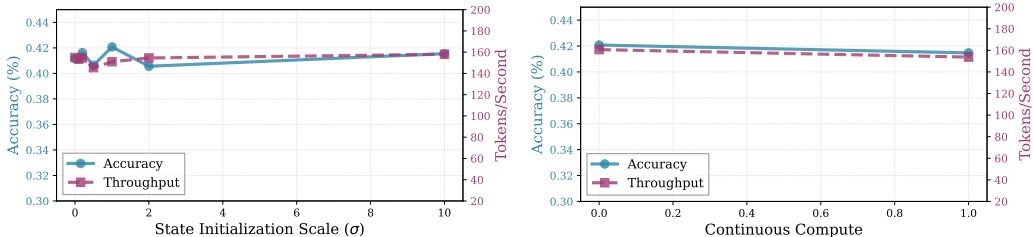

**Figure 10:** Impact of Additional Hyperparameter Choices, also on GSM8k. **Left** Initialization Scale of new states, which has only a minor effect of the result. **Right:** Continuous Compute, i.e. choosing to initialize new states with previously computed states (We initialize new states with the latest state from the position one step to the left). This is less effective for our sampler, given that the position one step to the left is only the result of $r'$ recurrences.

`claude-sonnet-4-5-20250929` to judge both completions. The model is given the system prompt and instruction for each query, and both responses, and is tasked to first reason about the quality of both responses (in line with the MTBench judge prompt), then rate both responses in terms of correctness, helpfulness, coherence and completeness on a scale of 1-10 and finally, to report whether it prefers the diffusion sampling response, or whether it ties the original. Aside from this evaluation, we also evaluate the log-perplexity of the generated completions, using either the original model, or an external model to judge perplexity. The external model we use is Qwen-3-8b-base. Also note that token/second in this chart are computed on a different accelerator than in other places in the paper, and as noted in the reproducibility statement, timing results are only to be compared within each figure or table.

### A.4 QUALITATIVE EVALUATION

To visualize the progress (or temporary lack thereof) of the sampler on a challenging sequence from the GSM8k validation set, we provide a few additional visualizations in Figure 13.

## B THEORETICAL ANALYSIS

### B.1 PROBLEM FORMULATIONS

**Definition B.1** (Depth and Width in Recurrent-Depth Models). Consider a recurrent-depth model $\mathcal{M}_d$ that processes an input sequence $\mathbf{x} \in \mathbb{R}^{L_0 \times h}$, where $L_0 \in \mathbb{N}$ is the sequence length and $h \in \mathbb{N}$ is the hidden dimension. At each generation step $t \in \mathbb{N}$, we define a *hidden state* as the $h$-dimensional output vector produced by a Transformer block for an input token. Let $\mathbf{H}_t \in \mathbb{R}^{w_t \times h}$ denote the 2D-matrix containing all hidden states at step $t$. We define the following two associated quantities:

- the *depth* $d_t \in \mathbb{N}$, defined as the number of *serial* Transformer block forward passes used to obtain $\mathbf{H}_t$ from the initial $L_0$ input tokens (i.e., the generation step), while ignoring any discretization;

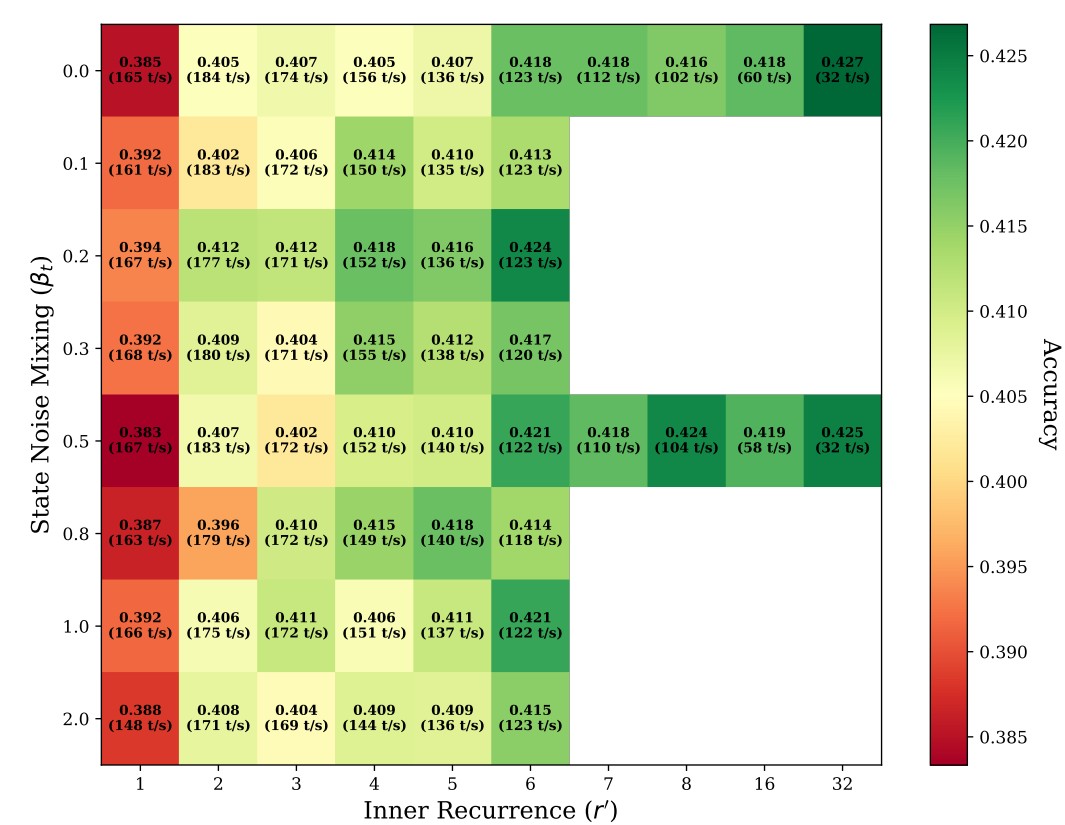

**Figure 11:** A heatmap of accuracy and throughput measurements spanned by varying noise and inner recurrence.

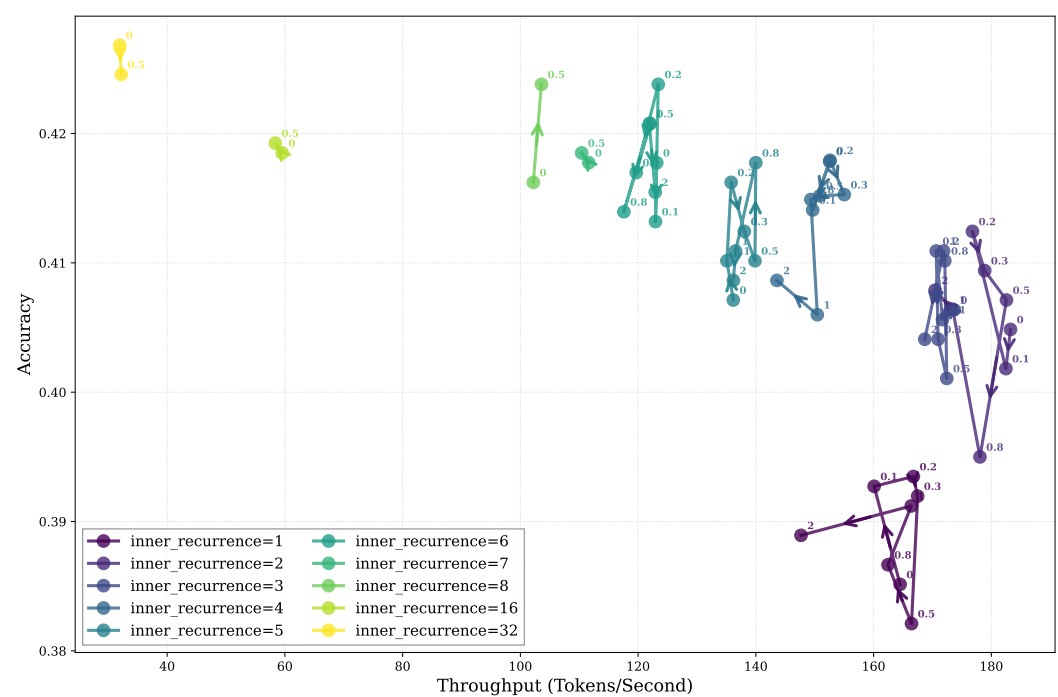

**Figure 12:** Additional visualizations of the trade-off of noise and inner recurrence in Figure 7.

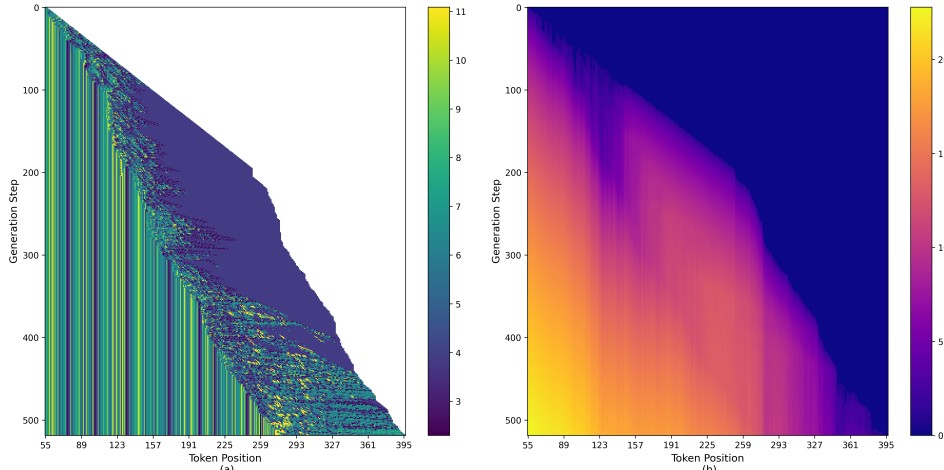

**Figure 13:** A full example of a sampler hyperparameter failure. As in Figure 4, this figure shows the token ids on the left, as they change during successive steps of the sampler (running from top to bottom) over the sequence dimension (running left to right). We see that the model tries various configurations for the current tokens, before they are gradually frozen as their latent states converge. Due to a few hard decisions (from the perspective of the model), early in the sequence, progress stalls until these tokens are decided, but then picks up speed again. However, large points of the wavefront all decode into the whitespace token (dark blue color), so that no useful states information is computed until the earlier tokens are resolved.

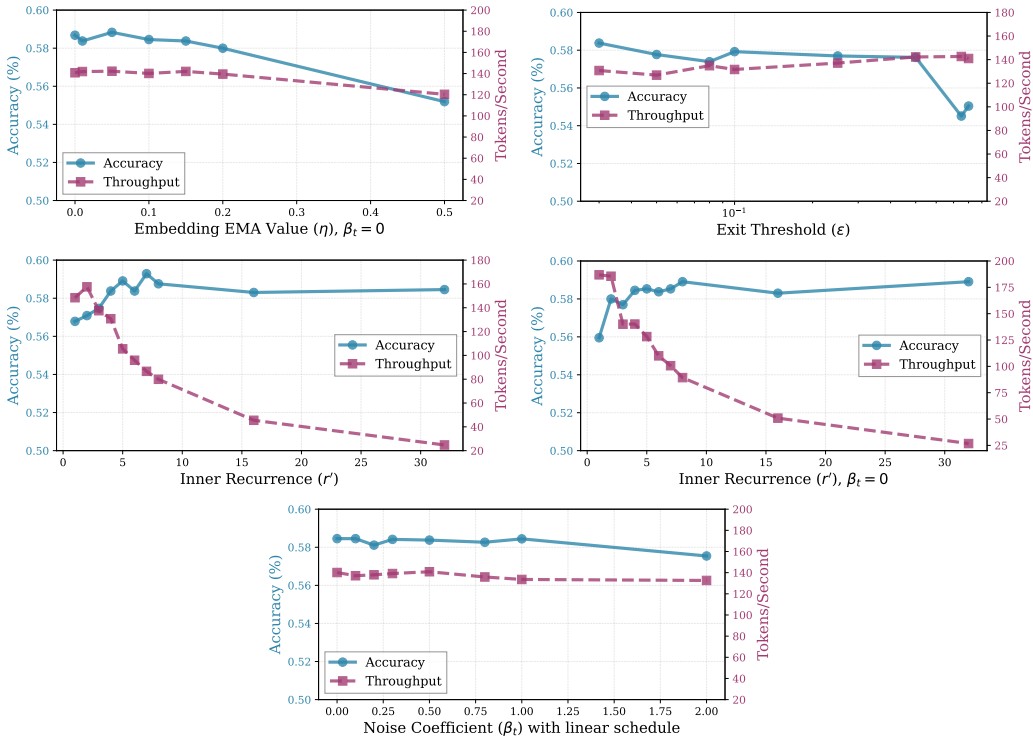

**Figure 14:** Hyperparameter Robustness for the finetuned math model on GSM8k. These figure repeat the ablation study from the main body concerning hyperparameter robustness also for the finetuned math model, showing that behaviors are largely similar, even though the model's capability has noticeably changed.

- the *width* $w_t \in \mathbb{N}$, defined as the cardinality of the active hidden-state set $\mathbf{H}_t$ ( i.e., the number of $h$-dimensional hidden states that are processed in *parallel* at generation step $t$).

These quantities evolve according to the following rules:

1. **Initialization.** At time $t = 0$, we set

$$d_0 = 0, \qquad w_0 = L_0.$$

2. **Depth update.** At each step $t \geq 0$, one additional Transformer block is applied, hence

$$d_{t+1} = d_t + 1,$$

so that $d_t = t$ for all $t \in \mathbb{N}$.

3. **Width update.** At each step $t \geq 0$, the width changes only due to two types of events:

   - *Token entry:* let $e^{(t)} \in \mathbb{N}_0$ denote the number of new tokens encoded into the model at step $t$, each contributing a new hidden state;
   - *Hidden-state exit:* let $x^{(t)} \in \mathbb{N}_0$ denote the number of hidden states removed from the model at step $t$ due to decoding.

   Then the width evolves as

   $$w_{t+1} = w_t + e^{(t)} - x^{(t)}.$$

   Equivalently, the net change can be written as $\delta^{(t)} = e^{(t)} - x^{(t)}$, so that $\delta^{(t)} > 0$ corresponds to entries (more tokens encoded), and $\delta^{(t)} < 0$ corresponds to exits (more hidden states decoded).

**Remark B.2.** *At any generation step $t$, all hidden states in $H_t$ share the same depth $d_t$, since each step corresponds to one additional serial forward pass through the Transformer block.*

### B.2 LLMs should prioritize depth scaling during prefilling.

**Definition B.3** (Width Scaling Variants)**.** Fix a width scaling factor $s \in \mathbb{N}$. Given an input sequence of length $L$, for each token $i \in \{1, \ldots, L\}$ we create $s$ copies indexed by $j \in \{1, \ldots, s\}$. The replicated sequence therefore has length $L \cdot s$, with elements denoted by $(i, j)$, the $j$-th copy of token $i$. The width-scaling model is obtained by applying a Transformer block (with parameters unchanged) to this replicated sequence under a customized attention mask, followed by a reduction step that maps the $L \cdot s$ outputs back to length $L$ (e.g., by selecting the last copy or averaging over copies).

We define two variants according to how each copy may attend:

- **Width-NoShare.** The $j$-th copy of token $i$ may attend to all copies of tokens $0, \ldots, i - 1$, as well as the first $j - 1$ copies of token $i$.

- **Width-KVShare.** The $j$-th copy of token $i$ may attend only to the last copy of tokens $0, \ldots, i - 1$, together with the first $j - 1$ copies of token $i$.

**Proposition B.4.** *During prefilling, both Width-NoShare and Width-KVShare are valid width-scaling architectures with factor $s$.*

*Proof.* **Depth.** At any generation step, each variant performs exactly one Transformer block forward pass on the replicated sequence. Therefore the number of serial block forward passes needed to produce the hidden states is unchanged, so the depth satisfies $\tilde{d}_t = d_t$.

**Width.** By definition, the width $w_t$ is the number of hidden states produced in parallel at step $t$. In the original model, prefilling a sequence of length $L$ produces $L$ hidden states per step. In both variants, we replicate each token $s$ times, so the block computes hidden states for all pairs $(i, j)$ with $i \in \{1, \ldots, L\}$ and $j \in \{1, \ldots, s\}$. Hence the total number of hidden states produced in that step is

$$\tilde{w}_t = Ls = s \cdot w_t.$$

The difference between NoShare and KVShare lies only in the attention pattern (which copies each query may attend to). This affects information flow but not the number of hidden states computed. The optional reduction back to length $L$ occurs *after* the parallel computation and thus does not change the measured width.

**Conclusion.** Both variants keep serial depth fixed and enlarge width by a factor of $s$, which is precisely our notion of width scaling. $\qquad\square$

