# OpenReview forum: "Efficient Parallel Samplers for Recurrent-Depth Models and Their Connection to Diffusion Language Models"
_ICLR.cc/2026/Conference — Submitted to ICLR 2026_

### Official Review · Reviewer_UbUX · 2025-10-31

**Soundness:** 2
**Presentation:** 4
**Contribution:** 3
**Rating:** 6
**Confidence:** 4

**Summary:**

This paper studies a sampler for a recurrent-depth Transformer language model by framing the recurrent-depth model as a diffusion model, motivated by a desire to increase sampling speed.

The authors detail the prerequisites of input injection and robustness to dynamic recurrence steps, as well as the importance of a recurrence-independent KV cache.

The sampler is proposed with either a fixed number of inner recurrences per-token, or with a simple adaptive exiting mechanism that emits final tokens when the normalized update distance is beneath a threshold.

The authors remark on the convergence of the proposed algorithm, as well as defend their sampler design with theoretical analysis of the costs of depth and width during prefilling and decoding respectively.

Finally, the authors demonstrate that the proposed sampler achieves similar accuracy on math and code generation to existing algorithms while processing tokens nearly 5 times faster.

**Strengths:**

- The proposed method elegantly decreases the time to sample from the studied recurrent-depth language model at apparently little-to-no decrease in performance
- The paper flows well and is cleanly presented
- This work introduces a potential connection between recurrent-depth language and diffusion models

**Weaknesses:**

- The evaluation is limited to one model family and solely evaluates math and coding reasoning, ignoring other recurrent-depth models and natural language tasks
- The evaluation excludes analysis of memory cost, as the model decreases wall-clock time by increasing compute parallelization on GPU
- Several remarks (Remark 3.1, Theorem 4.4, Conclusion), while thought provoking, may be misleading:
  - The assumption that the recurrent block is a contraction is difficult to believe
  - Theorem 4.4 states only the advantages (and not the disadvantages) of diffusion forcing sampling, and does not "prove that recurrent-depth models should use diffusion forcing samplers during decoding"
  - A diffusion-inspired sampling method does not indicate that the model being sampled from is a diffusion model

**Questions:**

- In some inference systems, computational resources may be limited at varying timepoints. Can you profile the memory usage over wall-clock time for the evaluated samplers?
- Figure 6 demonstrates that the noise coefficient strictly decreases performance, and is set to 0 in evaluations in Table 1 and Table 2. Is it actually beneficial? How unstable is the recurrence without the additive noise? Is sampling from the model quantifiably unstable?
- Most hyperparameter choices (momentum, noise coefficient, headway, maximum wavefront size, initialization scale, continuous compute) appear to have little-to-no trade-off, generally decreasing model performance. The connection to diffusion seems tenuous given this. Is the main motivation of the paper to introduce an efficient parallel sampler or to examine the relation between recurrent-depth models and diffusion language models?

---

> ### Author Response · Authors · 2025-11-21
> **Response to Reviewer UbUX (1/2)**
>
> Thank you for your interest in our work, careful comments and support of this submission. We’ll answer your questions one by one.
>
> ## Evaluation focused on one model family
>
> You are correct that our empirical evaluation is based on a single model family. To the best of our knowledge, this model family was the only option available at the time of submission that fit our constraints of an open-source model with downloadable weights, trained on generic language, of sufficient model size using recurrent depth with input injection. While this is potentially a limitation, we do think that most of our analysis is conceptual, and diffusion sampling can be applied to any recurrent-depth model following the ingredients outlined in Section 3.
>
> ## Adding an Analysis of Memory Costs
>
> This is a good suggestion. In practice, we implemented the maximum wavefront size parameter, see Fig.6, which bounds the maximum memory consumption, and then did measure memory as carefully. Based on your feedback we have now done so in a new Figure 8. This figure shows that the peak memory usage during inference is indeed a function of wavefront size. However, for the investigated model, the memory footprint of model states is relatively small, so that in the median, the diffusion sampler consumes about 250MB of additional memory (Fig 8 (left)).
>
> However, when tracking the peak memory usage over all samples in the dataset, we do see that stalls of the sampler (as shown in Fig.4 (right)) can increase memory costs, which are indeed bounded through the maximum wavefront size hyperparameter. For example, a maximum wavefront size of 64 constrains the practical peak memory added during generation for GSM8k to 3GB. Overall, we recommend picking the smallest maximum wavefront size that
> saturates throughput on a per-accelerator basis, to prevent worst-case memory allocation costs. It’s also interesting to consider that a more advanced inference engine/algorithm could dynamically schedule the maximum wavefront size to prevent memory peaks in situations where insufficient GPU memory is available.
>
>
> ## Remarks on Theorems and Remarks
>
> Thank you for your careful reading of our theorems and remarks. We are in agreement that the recurrent block is likely not a contraction - - we provide Remark 3.1 only as a conceptual guide to illustrate which condition would be sufficient to guarantee that both autoregressive and diffusion sampling coincide. Based on your feedback we have now changed the wording of the Remark slightly to clarify that this is indeed a hypothetical.
> Similarly, with Theorem 4.4 we cannot (at least not yet) fully characterize the disadvantages of diffusion forcing sampling. However, what we can say is that Theorem 4.4 describes the conditions under which one can hope for it to be beneficial at all. We further find the statement helpful, as it connects the various practical constraints of the sampler, such as wavefront size, and recurrence budget with the length of the target sequence. We agree and concede that this is likely not the last word on a theoretical understanding of how to optimally sample from recurrent-depth models, but we think these are positive first steps to understanding this topic.
>
> > A diffusion-inspired sampling method does not indicate that the model being sampled from is a diffusion model
>
> We agree! We have rewritten our conclusion statement regarding this sentence to more clearly describe this as a thought experiment.
>
> [our response continues in the next message]

---

> ### Author Response · Authors · 2025-11-21
> **Response to Reviewer UbUX (2/2)**
>
> ## Hyperparameter Choices
>
> > Figure 6 demonstrates that the noise coefficient strictly decreases performance, and is set to 0 in evaluations in Table 1 and Table 2. Is it actually beneficial? How unstable is the recurrence without the additive noise? Is sampling from the model quantifiably unstable?
>
> In the submission version, we (unfortunately) picked sub-optimal values for the other parameter ($r’$) for Fig.6 to show what difference $\beta_s$ makes. This was an oversight! **We have now rerun all ablations with additional runs and we have added a completely new figure, Fig.7 which now shows the full comparison of all combinations of inner recurrence and noise**.
>
> In Fig.7, we find a few interesting details. First, for most values of $r’$, choosing $\beta_s$ greater than zero is Pareto-optimal. Second though, the optimal $\beta_s$ values varies as a function of $r’$. The plot also confirms that for $r’=2$ (the original Fig.6), the value of $\beta_s=0$ is on the Pareto frontier. Yet all larger values of $\beta_s$ between 0.2 and 0.5 also lie on the frontier, and these values lead to more accurate solutions to GSM8k at only a minor cost in speed.
>
> Regarding your question, the recurrence is also usable without noise (which would be DDIM in our analogy to standard diffusion model samplers), which is marked in dark blue in Fig.7. However, looking at these points, we do actually find that $\beta_s$ values greater than zero achieve a better accuracy/speed trade-off (aside from the extreme of $r’=32$, where we are practically running the entire recurrence from the original paper before advancing to the next token, i.e. we are very close to the autoregressive approach).
>
>
> > hyperparameter choices (momentum, noise coefficient, headway, maximum wavefront size, initialization scale, continuous compute) appear to have little-to-no trade-off
>
> This is partially true - yet we do have to clarify that, based on our results, momentum, noise coefficient and maximum wavefront size are important hyperparameters (and we have run additional runs and repeated measurements to clarify).
>
> However, you are correct that we find that headway, initialization scale and continuous compute appear to have only little effects. We would argue that the fact that these hyperparameters are less important is a finding of our work that would not have been obvious beforehand. We considered not including these results, but considered that this would constitute hindsight bias, and that the robustness to these parameters is a finding.
>
> That being said, we have now switched around which of these we show in Figures 5 and 6 and which in the Appendix, as we do agree that the selection of which values to show in the main body was sub-optimal in the submission version. The other values are now in the Appendix.
>
> ----
>
> ## Summary
>
>
> In summary, we believe we were able to address at least your questions regarding memory costs and hyperparameter choices. Please let us know if this is not the case. We’d be very happy to continue this conversation, including new questions or other concerns!

---

> > ### Comment · Reviewer_UbUX · 2025-11-23
> >
> > I think that the first line of Section 4.3, line 399
> > > Next, we **prove** that recurrent-depth models should use diffusion forcing samplers during decoding.
> > should be changed.
> >
> > This section contains an argument, not a formal proof.
> >
> > Otherwise I am happy with this work and raise my score.

---

> ### Author Response · Authors · 2025-12-03
>
> Dear Reviewer UbUX, thank you for your continued interest and support of our submission, and for raising your score to 8.
>
> Regarding your final comment, concerning the wording choice in the introduction sentence of Section 4.3, thank you for bringing this up! We have now updated our PDF to be more precise about our argument in Section 4.3.

---

### Official Review · Reviewer_5EMW · 2025-11-02

**Soundness:** 3
**Presentation:** 2
**Contribution:** 2
**Rating:** 4
**Confidence:** 4

**Summary:**

This paper proposes an efficient mechanism for parallelizing the extra computation in recurrent-depth language models to accelerate their slow inference speed. Specifically, it introduces a diffusion forcing sampler that, at every forward pass, decodes a new "draft" token at the end of the sequence while simultaneously refining the latent states of all previously drafted tokens in parallel. This sampler uses an adaptive exit criterion to "freeze" tokens once their latent states stabilize, allowing the generation to proceed in an efficient "wave". The experiments show that this sampler can be applied directly to existing 3.5B recurrent-depth models without any retraining, leading to a 5x speedup on reasoning and coding benchmarks with only minor trade-offs in accuracy.

**Strengths:**

1. The connection between recurrent latent update and diffusion sampling is relevant and interesting.
2. The proposed sampler achieves significant speedup with only mild degradation of quality.
3. The method is designed to work with KV cache sharing, which allows it to have a memory footprint no larger than a standard fixed-depth transformer, preventing the cache from growing with the number of recurrence steps.
4. The authors explored some key factors that affects the stability of latent recurrence.

**Weaknesses:**

1. The theoretical analysis section seems not fully formalized. The concepts of "depth scaling" and "width scaling" seem to be created specifically for their argument, rather than established, well-understood principles. Furthermore, the paper's core mechanism relies on the recurrent-depth model's states converging. However, the authors admit in Remark 3.1 that they cannot formally prove this. Sec 4.2 seem disconnected from the other part of the paper. There could be simpler way to formalize the high level intuitions.
2. The authors explicitly state that their experimental evaluation is limited to a batch size of 1. They acknowledge that extending the sampler to batched or continuously-batched inference is complex and "fall outside the scope of this study". This is a limitation but I look forward to future development of the proposed method.
3. In Appx A.2, the authors state that "We experiment with headways greater than one, but while interestingly stable, this accelerates the speed of the sampler only slightly, at a large cost to accuracy". So even though the paper is written in a way that strongly relates to diffusion models, the generation is still largely next-token prediction.

**Questions:**

Why does the noise injection to z (Eq. 2) only happen at the start of a new round of inner recurrence? Actually, is it helpful or not? It looks like in the reported results, $\beta_s$ is set to 0. And Fig. 6 shows that on the GSM8k benchmark is achieved at $\beta_s = 0.00$. What motivates the introduction of this $\beta_s$ apart from connecting it to Diffusion forcing? And how to understand its impact on the throughput?

---

> ### Author Response · Authors · 2025-11-21
> **Response to Reviewer 5EMW (1/2)**
>
> Thank you for your careful feedback. We’re glad you found the connection between latent recurrence and diffusion sampling interesting! Regarding your questions:
>
> ## Questions regarding Section 4 (Theoretical Analysis)
>
> > The concepts of "depth scaling" and "width scaling" seem to be created specifically for their argument, rather than established, well-understood principles.
>
> To some extent, yes, you are correct, these are indeed concepts we are trying to formalize for the first time, in this work, and this section. We are trying here to establish a language to think about these types of scaling. This is not well established, as “normal” fixed-depth transformers cannot adapt their depth through recurrence, their depth is always fixed.
>
> The concept of width scaling is also relatively new, and, for example, would not apply to the original recurrent-depth model of Geiping2025, which always generates tokens with a width of 1. However, we do base our framework in turn on the concept of width scaling established in Wu et al. 2025 (“Efficient Pretraining Length Scaling”), who formalize this concept for fixed-depth transformers with KV-cache sharing. Section 4.3 in that paper, titled “Latent Thinking Transformers” clarifies this connection, speculating on a combination of scaling both width and depth.
>
> In this prior work, the authors still mention that “However, recurrent models
> cannot be used in parallel generation, leading to limited efficiency during inference.” In our new submission, we resolve this open question, creating models that allow for both depth and width scaling. We certainly are open to suggestions on how to tighten our framing of these concepts, as we created our theoretical results from scratch to better understand the novel computational trade-offs between these two particular forms of scaling.
>
> > Furthermore, the paper's core mechanism relies on the recurrent-depth model's states converging.
>
> While convergence of all latent states is a theoretically sufficient criterion to guarantee the equivalence between the discussed diffusion sampling strategy and standard sampling, we bring this up only to clarify the mental model of the reader. We do not expect the sequences generated by the diffusion sampler to be strictly equivalent in practice, and measure the accuracy of sequences generated through diffusion sampling in all figures and tables, to make the *empirical* argument that for many settings of the sampler, both auto-regressive and diffusion-forcing generation produce results that are similar in quality. We have modified the remark slightly to clarify this.
>
>
> ## Questions regarding bigger batch sizes
> > extending the sampler to batched or continuously-batched inference is complex and "fall outside the scope of this study". This is a limitation but I look forward to future development of the proposed method
>
> Thank you for your understanding on this point. With this submission, we wanted to focus on the principles of the connection between recurrence and diffusion, and think a proper inference engine that schedules, depth-scaling, width-scaling and batches would deserve its own dedicated research project, as we discuss in Appendix A.2.
>
>
> ## What does it mean for the sampler to use a headway of 1
>
> > We experiment with headways greater than one, but while interestingly stable, this accelerates the speed of the sampler only slightly [...]  So even though the paper is written in a way that strongly relates to diffusion models, the generation is still largely next-token prediction.
>
> Ah, we’re sorry to have caused this misunderstanding. It is important to separate the “headway” of the sampler, with which we denote the movement (width scaling) of the current subset of tokens that are “diffused/denoised”, from the “wavefront size” of the diffusion sampler. You are correct that a wavefront size of 1 would correspond to next-token prediction. However, that a headway of 1 is optimal only corresponds to the speed with which the wave of currently-changing positions is moving. At each step of the sampler, we add one token position to our pool of “active” tokens, from which tokens only leave once they are “frozen”.
>
> This is best exemplified by Figure 4 (middle). Each horizontal slice through Fig.4 is one step of the sampler. To the left are tokens that are frozen and hence unchanged between steps of the sampler, whereas up to 64 (=max_wavefront) tokens are being modified simultaneously in each step through diffusion forcing.
>
>
> [Our response continues below]

---

> ### Author Response · Authors · 2025-11-21
> **Response to Reviewer 5EMW (2/2)**
>
> ## Questions regarding noise injection and other hyperparameters
>
> > Why does the noise injection to z (Eq. 2) only happen at the start of a new round of inner recurrence? Actually, is it helpful or not?
>
> At each step and so at the start of each new round, the currently predicted tokens are re-embedded into $e$. As such, this is the point where the current state $s$ could be mismatched with the embedded inputs. Intuitively speaking, the noise injection resets the state and allows it to re-adapt to the changed embeddings with the inner recurrence. This is in line with established procedures to adapt diffusion models to changes in the conditioning, e.g. in “universal guidance for diffusion models” (Bansal et al. 2023), Section 3.3.
>
> > Fig. 6 shows that on the GSM8k benchmark is achieved at $\beta_s=0.0$
>
> In the submission version, we (unfortunately) happened to pick sub-optimal values for the other parameter ($r’$) for Fig.6 to show what difference $\beta_s$ makes. This was an oversight! Based on your feedback to better show the effect of $\beta_s$ **we have now added a new Figure, Fig.7, on the 10th page of the submission, showing the full Pareto frontier of inner recurrence and noise parameters**.
>
> In Fig.7, we find several interesting details. First, for most values of $r’$, choosing $\beta_s$ greater than zero is Pareto-optimal. Second, though, the optimal $\beta_s$ values vary as a function of $r’$. The plot also confirms that for $r’=2$ (the original Fig.6), the value of $\beta_s=0$ is on the Pareto frontier. Yet all larger values of $\beta_s$ between 0.2 and 0.5 also lie on the frontier, and these values lead to more accurate solutions to GSM8k at only a minor cost in speed.
>
> ---
> ## Summary
> Overall, thank you for your careful reading of our work! We think we were able to address your key questions regarding the origin of the concepts of depth/width scaling, and the hyperparameter choices of headway and noise. Please let us know if this is not the case, if there are any further questions, or if you have any comments or follow-up thoughts. We’d be happy to continue this discussion.

---

### Official Review · Reviewer_EH9x · 2025-11-08

**Soundness:** 2
**Presentation:** 3
**Contribution:** 2
**Rating:** 2
**Confidence:** 3

**Summary:**

This paper proposes parallelizing inference for recurrent-depth transformers by processing multiple token positions simultaneously at different recurrence depths, achieving significant speedup. The approach is claimed to connect recurrent-depth models to diffusion models and can be applied to existing models without retraining.

**Strengths:**

* This paper tried to address a genuine bottleneck in recurrent-depth model inference, and the acceleration is significant.

* This method makes a good approach that can be applied directly to existing models without further training.

**Weaknesses:**

* The base model that this paper used was not proved to be fundamental better than GPT/AR based method. The contribution of this paper is questioned given if the recurrent-depth model is a promising direction or not.

* The method cannot guarantee producing the same output as sequential generation, which creating fundamentally different computational paths. For use cases requiring reproducibility, this is a non-starter. The paper should clearly state this limitation and specify when the method is/isn't appropriate.

* The evaluation results are very limited. All the four benchmarks share a very particular property: they use extraction-based metrics that completely ignore generation quality. What completely missing are summarization, translation, long-form QA, dialogue, creative writing, and general language understanding tasks where fluent generation matters. More language quality metrics are reported (perplexity, BLEU, human evaluation, token-level accuracy), while there are 15+ benchmarks results in the base model.

* I'm not an expert of diffusion, but I think the connection to diffusion models is superficial to the point of being misleading. The paper reveals noise was added post-hoc as a hack, not as part of a principled diffusion framework.

**Questions:**

* Why only four benchmarks when the base model likely tested on 10-15? Can you provide results on all benchmarks from Geiping et al. 2025, particularly summarization, translation, long-form generation, dialogue, and general language understanding tasks?

* Can you provide perplexity, BLEU/ROUGE scores, human evaluation, and token-level accuracy to assess actual generation quality rather than just final answer correctness?

* What specifically makes this "diffusion" versus standard iterative refinement? Was the model trained with any diffusion objective, or is the connection purely post-hoc?

* On which task types does the method degrade significantly? Are there examples where generation quality is poor despite correct final answers?

---

> ### Author Response · Authors · 2025-11-21
> **Response to Reviewer EH9x (1/2)**
>
> Dear reviewer, thank you for your feedback! We will address your questions below:
>
>
> ## Benchmarking Questions
>
>
> > More language quality metrics are reported (perplexity, BLEU, human evaluation, token-level accuracy), while there are 15+ benchmarks results in the base model.
>
> > Why only four benchmarks when the base model likely tested on 10-15? Can you provide results on all benchmarks from Geiping et al. 2025, particularly summarization, translation, long-form generation, dialogue, and general language understanding tasks?
>
> This is (unfortunately) a hallucination, the referenced paper (Geiping et al. 2025) does not contain benchmarks on ‘summarization, translation, long-form generation, dialogue’.
>
> The referenced paper which introduces the base model does contain 8 base model benchmarks (ARC-easy, ARC-challenge, HellaSwag, MMLU, openbookQA, PiQA, SciQ and Winogrande) used to evaluate natural language understanding. However, these benchmarks are not generative, they are evaluated as cloze tests (where the base model probabilities of different answers are evaluated). As such, no answers are generated and the sampling strategy we discuss to *parallelize generation* does not apply (or alternatively: All benchmark scores will be exactly the same). Our submission evaluates *all* generative benchmarks from Geiping2025, so we do indeed cover all benchmarks from that work!
>
> You do have a point that we could have evaluated more generative benchmarks in this work to track the fluency of generations produced by the diffusion sampler. We did not do this for the submission because our main worry was that the diffusion sampling would break on complex reasoning problems such as GSM8k.
>
> Based on your feedback **we have now added a completely new set of experiments to page 10 (Table 3), where we evaluate the generation quality of free-form responses to four prompt datasets (AlpacaEval, LitBench, MTBench and Hermes-3) based on 10 metrics (ROUGE-1, ROUGE-L, Response Overlap, Judge Preference, Coherence, Helpfulness, Correctness, Completeness, self-perplexity, perplexity on reference models)**. Comparing autoregressive sampling and the diffusion sampler on these benchmarks, we find that the trade-off between performance and accuracy observed in free-form generations is similar to the trade-off observed for reasoning tasks like GSM8k and HumanEval.
>
> > I'm not an expert of diffusion, but I think the connection to diffusion models is superficial
>
> In this work, we have focused on a self-contained explanation of how to apply diffusion forcing sampling, which we reference as the analogue for diffusion models. But, this analogy is quite apparent, for example when comparing with the original diffusion paper (https://arxiv.org/abs/2407.01392), e.g. in Fig.2 (“Method overview”). In our outline of Section 3, we introduce elements of a diffusion forcing sampler one by one, and so we also introduce the addition of noise separately, but this addition is directly informed by the noise scheduling described in Section 3.2 of https://arxiv.org/abs/2407.01392, see paragraph “Sampling”).
>
> > What specifically makes this "diffusion" versus standard iterative refinement? Was the model trained with any diffusion objective
>
> What makes this sampling strategy “diffusion” is the direct use of diffusion forcing sampling. With this work we highlight this surprising conceptual connection that recurrent-depth models, which are emphatically *not* trained with a diffusion objective, can be treated at inference just like latent continuous diffusion models (LDMs). We make this connection in Section 3.1, connecting the input injection of recurrent-depth models to the continuous conditioning in LDMs, and the randomized state initialization is equivalent to the randomized latent space initialization of the solution in an LDM.
>
> [Our response continue in Response 2]

---

> ### Author Response · Authors · 2025-11-21
> **Response to Reviewer EH9x (2/2)**
>
> ## Miscellaneous Questions
>
> > Are there examples where generation quality is poor despite correct final answers?
>
> You can see suboptimal solutions in the latent space of the sampler in Fig. 4 (middle+right). There, we find that with suboptimal hyperparameters, the sampler stalls in latent space and cannot advance until earlier tokens are resolved. This is a failure case for generation, as it negatively impacts the speed of the sampler.
>
> > The method cannot guarantee producing the same output as sequential generation, which creating fundamentally different computational paths.
>
> Yes, this is a cornerstone of our parallelization strategy (see Fig. 1). We have modified our introduction to give additional emphasis to this fact. We think there should be no confusion now.
>
> > The base model that this paper used was not proved to be fundamental better than GPT/AR based method. The contribution of this paper is questioned given if the recurrent-depth model is a promising direction or not.
>
> We do think this is a bit of a narrow view on the scientific method. Many promising new architectures (indeed even the transformers paradigm itself) were only developed through a series of papers in conferences such as ICLR. With this work, we show that one disadvantage of recurrent-depth transformers, namely their slow inference speed, can be effectively solved through parallelization, therefore opening up new  advantages for these architectures.
>
> Concerns about comparative advantage between architectures may not age as well. In particular, note that, this new paper: https://arxiv.org/abs/2510.25741 has very recently shown that recurrent-depth models can catch up to strong open-source models, if only trained with similar compute budgets.
>
> ----
> ## Summary
>
> Overall, thank you again for your review comments. We’re glad we were able to clarify that we do run all benchmarks from Geiping2025 where the outcome could change when the sampler is changed. Based on your feedback we have now also added *40 new results* measuring the quality of free-form generations in varied settings, where we explicitly pick up ROUGE and perplexity metrics, as mentioned in your review.
>
> Please let us know whether these results resolve your concerns. We’d be happy to provide additional details, or run additional free-form datasets.

---

### Author Response · Authors · 2025-12-04
**General Summary**

Dear readers, reviewers, old AC and new ACs,

Thank you for your sustained engagement with our submission. We’ll briefly recap our contributions, reviewer feedback and how we addressed feedback during the review process.

## Our Contributions
Several works this year have shown that language models that recur in depth (i.e. repeat their layers) for every token may be promising architectures for reasoning. However, generating text from these models is generally considered to be bottlenecked by the sequential structure of recurrence: for every token generated, the full recurrence has to be computed before the next token can be generated.

In this work, we show for the first time that this is not a fundamental constraint: Generation from recurrent-depth models can be parallelized in the sequence dimension by analogy to diffusion forcing (see Fig.1 in the submission). This concept derived through analogy with diffusion modeling is our core contribution.

We then show that this type of diffusion sampler can also be applied in practice to accelerate the generation speed of the largest recurrent-depth language model available at submission, accelerating the generation speed by 4-5x, even on challenging reasoning benchmarks with only minor trade-offs in performance of about 1%, without any modifications of the underlying model.

## Reviewer Feedback and Our Updates

Reviewers brought up a few key questions:
1.   How does this new sampling strategy impact text quality, would the sampler also work in domains such as creative writing, how would it impact the model’s free-form generation quality in chat domains (*reviewer EH9x*).

*Our Update*: To address this question, we added 40 new measurements of qualitative metrics of text quality, measured through standard metrics (ROUGE, perplexity) as well as LLM-as-a-judge preference comparisons and rating. For four datasets and each metric we compare the answer generated by the baseline, autoregressive sampler, and the answer generated by the diffusion sampler.

2.    What is the optimal value for noise added during generation with the parallel sampler, is noise addition actually optimal? (*reviewers UbUX, 5EMW*)

*Our Update*: We have added a new figure (Fig.7), where we now plot the full Pareto frontier of accuracy vs throughput as a function of our hyperparameters of inner recurrence and noise scale. With this, we can now show the full picture, detailing for which levels of recurrence it is optimal to add noise. We show that for intermediate levels of recurrence, adding noise does increase throughput significantly.

3.   What is the memory footprint of the parallelized diffusion sampler? (*reviewer UbUX*)

*Our Update*: We have added a new Section (A.2) to extensively describe the memory characteristics of the sampler. The new, corresponding Fig.8 now shows the footprint of the sampler in both the median case and the worst-case. Memory requirements of the sampler are bounded by the maximum size of the sampler wavefront (as introduced in Section 3.5), which is now also supported by the measurements shown in Fig.8.

----

We thank all reviewers for their engagement and questions, and we are glad that we were able to address these key questions.

---

### Meta-Review · Area_Chair_Xe1s · 2026-01-07

**Summary:**

This paper proposes a parallel sampler for recurrent depth LMs by drawing connections to diffusion models. The paper proposes a set of theoretical justifications, and demonstrates up to 5x speedup when evaluated on a 3.5B recurrent depth Transformer.

The paper received mixed reviews. All reviewers acknowledges it as a sound approach addressing a valid problem, and experimental results are significant. However, concerns are raised in terms of its framing/claims, as well as completeness of evaluations. The rebuttal addresses some of the concerns, but the AC believes that there are more remaining. Especially, the connection between the proposed approach and diffusion seems like a stretch and to some degree misleading/unnecessary, as pointed out by Reviewer EH9x and UbUX. Another consideration is that recurrent depth Transformers, though a valid line of research, is a niche area which makes the scope of this work pretty narrow. Based on these factors, the AC cannot recommend accept for this round but would encourage the authors to continue improving it.

**Reviewer Concerns:**

The most significant outstanding concerns are the unnatural connection to diffusion, as well as related claims regarding theoretical justifications (eg convergence and recurrent depth being a contraction). Refer to the summary for more.

**Reviewer Scores:**

UbUX has indicated that they were willing to increase the score; Some of the concerns of EH9x on evaluations are also effectively addressed by the rebuttal so I would reasonably expect an increase in score as well, but probably still below the acceptance threshold.

---

### Decision · Program_Chairs · 2026-01-26

Reject